# BAYESIAN INCREMENTAL LEARNING FOR DEEP NEURAL NETWORKS

**Max Kochurov**[1,2], **Timur Garipov**[1,2], **Dmitry Podoprikhin**[2]
**Dmitry Molchanov**[3], **Arsenii Ashukha**[1,4], **Dmitry Vetrov**[1,3]
[1]AI Centre, Samsung Research Russia, [2]Moscow State University,
[3]National Research University Higher School of Economics, [4]University of Amsterdam
{maxim.v.kochurov, timgaripov, timmyofmexico}@gmail.com
{dmolch111, ars.ashuha, vetrodim}@gmail.com

## ABSTRACT

In industrial machine learning pipelines, data often arrive in parts. Particularly in the case of deep neural networks, it may be too expensive to train the model from scratch each time, so one would rather use a previously learned model and the new data to improve performance. However, deep neural networks are prone to getting stuck in a suboptimal solution when trained on only new data as compared to the full dataset. Our work focuses on a continuous learning setup where the task is always the same and new parts of data arrive sequentially. We apply a Bayesian approach to update the posterior approximation with each new piece of data and find this method to outperform the traditional approach in our experiments.

## 1 BAYESIAN INCREMENTAL LEARNING

Recent work has shown promise in incremental learning; for example, a set of reinforcement learning problems have been successively solved by a single model with a help of weight consolidation (Kirkpatrick et al., 2016) or Bayesian inference (Nguyen et al., 2017). In this work we focus on a specific incremental learning setting – we consider a single fixed task when independent data portions arrive sequentially. We formulate a Bayesian method for incremental learning and use recent advances in approximate Bayesian inference (Kingma & Welling, 2013; Kingma et al., 2015; Louizos & Welling, 2017) to obtain a scalable learning algorithm. We demonstrate the performance of our method on MNIST and CIFAR-10 is improved relative to a naive fine-tuning approach and can be applied to a conventional (non-Bayesian) pre-trained DNN.

Consider an i.i.d. dataset $\mathcal{D} = \{x_i, y_i\}_{i=1}^N$. In an incremental learning setting, this dataset is divided into $T$ parts $\mathcal{D} = \{\mathcal{D}_1, \ldots, \mathcal{D}_T\}$, which arrive sequentially during training. The goal is to build an efficient algorithm that takes a model, trained on the first $t-1$ units of data $\mathcal{D}_1, \ldots, \mathcal{D}_{t-1}$, and retrain it on a new unit of data $\mathcal{D}_t$ without access to $\mathcal{D}_1, \ldots, \mathcal{D}_{t-1}$ and without forgetting dependencies.

The most naive deep learning approach for incremental learning is to apply the Stochastic Gradient Descent (SGD) updates with the same loss function on the new data parts, to *fine-tune* the model. However, in that case, the model is likely to converge to a local optima on a new data unit without saving the information learned from the previous parts of the data.

The Bayesian framework is a powerful tool for working with probabilistic models. It allows to estimate the posterior distribution $p(w \,|\, \mathcal{D}_1, \ldots, \mathcal{D}_t)$ over the weights $w$ of the model. We can use the Bayes rule to sequentially update the posterior distribution in the incremental learning setting:

$$p(w \,|\, \mathcal{D}_1, \ldots, \mathcal{D}_t) \propto p(\mathcal{D}_t \,|\, w) p(w \,|\, \mathcal{D}_1, \ldots, \mathcal{D}_{t-1}) \qquad (1)$$

Unfortunately, in most cases the posterior distribution $p(w \,|\, \mathcal{D}_1, \ldots, \mathcal{D}_t)$ is intractable, so we can use *stochastic variational inference* (Hoffman et al., 2012) to approximate it. In the next section we present a scalable method for incremental learning, and study different variational approximations of the posterior distribution.

## 2 SCALABLE METHOD FOR BAYESIAN INCREMENTAL LEARNING

We apply variational inference to approximate $p(w \,|\, \mathcal{D}_1, \ldots, \mathcal{D}_t) \approx q(w \,|\, \phi_t) \in \mathcal{Q}$ with access only to $D_t$ and the previous approximation $q(w \,|\, \phi_{t-1})$. To train our model we follow Kingma & Welling (2013) and use the reparameterization trick to obtain an unbiased differentiable minibatch-based Monte Carlo estimator of the variational lower bound

$$\mathcal{L} = \mathbb{E}_{q(w \,|\, \phi_t)} \log p(\mathcal{D}_t \,|\, w) - D_{\mathrm{KL}}(q(w \,|\, \phi_t) \,||\, p_t(w)) \to \max_{\phi_t} \qquad (2)$$

The prior distribution $p_t(w)$ is the posterior approximation $q(w \,|\, \phi_{t-1})$ from the previous step. Unfortunately, this approximation is not exact and as a result, the incremental procedure becomes biased. The quality of the incremental learning algorithm depends strongly on the posterior approximation $q$, with more expressive families having a lower approximation gap, but poorer stability. We investigate, how different approximations behave in a Bayesian incremental learning algorithm.

**Fully Factorized Gaussian Approximation** is a fast, stable and easy to use approximation family. For a dense layer with input and output dimension $I$, $O$, respectively, the model is:

$$q_\phi(w) = \prod_{i=1}^{I} \prod_{j=1}^{O} \mathcal{N}(w_{ij} \,|\, \mu_{ij}, \sigma_{ij}^2). \qquad (3)$$

The approximate posterior for a convolutional layer factorizes similarly over all kernel parameters. Gaussian approximation is widely used (Blundell et al., 2015; Kingma et al., 2015; Kucukelbir et al., 2015; Molchanov et al., 2017), however this family has low expressiveness (Louizos & Welling, 2017) which affects the quality of incremental learning.

Next, we consider a convolutional layer with $N$ filters and $C$ channels with filter size $H \times W$.

**Channel Factorized Gaussian Approximation** nicely fits convolutional layers preserving correlations within kernel parameters channel-wise. Following Rezende et al. (2014) we use the Cholesky decomposition to parameterize the covariance matrix. Under this parameterization, we can both perform the reparameterization trick and compute the density efficiently.

$$q_\phi(w) = \prod_{n=1}^{N} \prod_{c=1}^{C} \mathcal{N}(w_{nc} \,|\, \mu_{nc}, L_{nc} L_{nc}^\top), \qquad (4)$$

where $L_{nc} \in \mathcal{R}^{HW \times HW}$ denotes a lower triangular matrix with positive diagonal elements.

**Multiplicative Normalizing Flow Approximation** is a highly expressive variational family. Louizos & Welling (2017) successfully employed it to train Bayesian deep neural networks. MNFs introduce an auxiliary variable $z$ to define a posterior approximation $q(w \,|\, \phi)$:

$$q_\phi(w \,|\, z) = \prod_{n=1}^{N} \prod_{c=1}^{C} \prod_{i=1}^{H} \prod_{j=1}^{W} \mathcal{N}(w_{ncij} \,|\, z_{nc}\mu_{ncij}, \sigma_{ncij}^2), \quad z = NF(z_0), \qquad (5)$$

where $z_0$ follows simple distribution $q(z_0)$ and $NF$ is a normalizing flow (Rezende & Mohamed, 2015). However, this approximation can not be used in incremental learning because it requires computing an intractable integral to calculate $q(w \,|\, \phi)$ (a prior on the next incremental step). To address this issue, we derive a new variational lower bound (Appendix C) to optimize the joint approximation $q(w, z \,|\, \phi)$, instead of the marginal $q(w \,|\, \phi_t)$:

$$\mathcal{L} = \mathbb{E}_{q(w, z \,|\, \phi_t)} \log p(\mathcal{D}_t \,|\, w) - D_{\mathrm{KL}}(q(w, z \,|\, \phi_t) \,||\, q(w, z \,|\, \phi_{t-1})) \qquad (6)$$

The key difference relative to the original lower bound is that we treat $z$ as a regular parameter and not as an auxiliary variable. This allows us to use a joint prior $p_t(w, z)$, which leads to a tractable incremental learning procedure. We hope that a more complex posterior will help in an incremental setting.

**Pretraining**. When training a large neural network, it is beneficial to use a model that has been *pre-trained* on another task for initialization. In order to apply the Bayesian approach, one would need to specify a prior distribution over the weights given the pretrained DNN. The simplest choice is a fully-factorized Gaussian prior $p(w) = \mathcal{N}(w \,|\, w^\star, \sigma^2)$ centered around the pretrained value $w^\star$ and with some fixed variance $\sigma^2$. Typically, one would use grid search for $\sigma^2$, but a better approach might be to use the Laplace approximation (Azevedo-filho, 1994) to obtain $\sigma^2$ given old data. Fitting a Laplace approximation, we obtain individual $\sigma^2$ for every weight, which appears to be beneficial based upon our experiments.

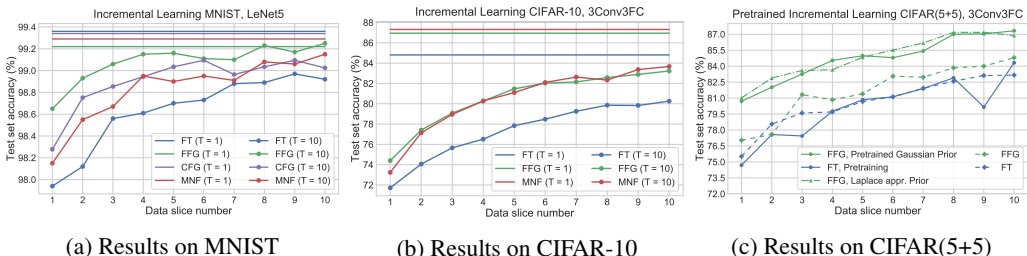

(a) Results on MNIST            (b) Results on CIFAR-10            (c) Results on CIFAR(5+5)

Figure 1: Accuracy in Incremental Learning setting on MNIST, CIFAR-10, CIFAR(5+5) datasets. We denote Fine Tuning as FT, Fully Factorized Gaussian as FFG, Channel Factorized Gaussian as CFG, Multiplicative Normalizing Flows as MNF. We present a comparison of the conventional fine-tuning and the proposed Bayesian incremental learning approach on MNIST (a) and CIFAR-10 (b) datasets. The accuracy for the test set is reported for different posterior approximations, both for the training on the full dataset ($T$=1) and for the incremental learning of the dataset divided into $T$=10 parts. The experiment demonstrates that the proposed approach allows us to outperform classical fine-tuning approach. Bayesian Incremental Learning results for pretrained and non-pretrained networks on the CIFAR(5+5) task is presented at (c). We reported accuracy values in incremental learning setting on the subset of the test set which corresponds to the second stage in CIFAR(5+5) experiment.

## 3 EXPERIMENTS

In our experiments, we compared test set accuracy after incremental training on MNIST and CIFAR-10 datasets for LeNet5 (LeCun et al., 1998) and 3Conv3FC (Hinton et al., 2012) architectures, respectively, using the proposed approach and fine-tuning. Details for the training procedure are described in Appendix A.

**Incremental Learning on MNIST and CIFAR-10** The fine-tuning (FT) approach achieved the same score in a non-incremental setting (with $T$=1) compared to Bayesian methods 1. However, it failed to solve the incremental learning task ($T$=10), resulting in low classification performance. Fully-factorized Gaussian approximation (FFG) solves the problem successively and matches the performance of a non-incremental setting ($T$=1). Normalizing flows (MNF) performed worse than the fully-factorized Gaussian approximation in the incremental learning setting, but better when $T$=1. We expect the optimization gap is due to few data available and unstable convergence of complex posteriors.

We have experienced optimization problems when moving to larger architectures. In larger architectures, the data term is being dominated by the KL-term in the objective, which leads to severe underfitting. To cope with this problem, we downscaled the KL term by $0.05$, which is a common trick in Bayesian deep learning (Ullrich et al., 2017; Higgins et al., 2017). The resulting objective is no longer a proper variational lower bound, but it works well in practice and outperforms fine-tuning by a large margin.

**Incremental Leaning with Domain Adaptation on CIFAR(5+5)** In this experiment we evaluated the performance of pretraining approaches using CIFAR(5+5) dataset we got by randomly dividing CIFAR-10 into two equal parts based on labels. Pretraining was done on the first half of the data while incremental task was solved on the second half. We compared grid search and the Laplace approximation for $\sigma^2$ described above to the fine-tuning approach. Experiments showed that pretraining helps to improve the performance of Bayesian neural networks trained incrementally on the rest of data. However, the fine-tuning approach fails to benefit from pre-training. Moreover, the experiments showed that the Laplace approximation performs well, so the grid search is not necessary.

ACKNOWLEDGMENTS

This research was supported by Samsung Research, Samsung Electronics.

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

APPENDIX

A    BAYESIAN INCREMENTAL LEARNING ALGOTIRHM

---

**Algorithm 1** The Bayesian Incremental Learning by DSVI

---
1: **procedure** BAYESIANIL($q_\phi, \mathcal{D}^{iter}$)                    ▷ Var Approximation, Data Iterator
2:    $\phi_0 \leftarrow$ Initialize parameters of prior distribution
3:    $\phi_1 \leftarrow$ Initialize variational parameters
4:    **for** $t \in [1, \ldots, T]$ **do**                    ▷ New data-part arrives sequentially
5:       **repeat**
6:          $\mathcal{D}^M \leftarrow$ Minibatch of M datapoints drawn from dataset $\mathcal{D}_t$
7:          $g \leftarrow \nabla_{\phi_t} \mathcal{L}$ from (2)
8:          $\phi_t \leftarrow$ Update parameters using g (e.g. SGD or Adam)
9:       **until** convergence of parameters $\phi_t$
10:       $\phi_{t+t} \leftarrow \phi_t$ Initialization for the next iteration
11:    **return** $q(w; \phi_T)$                    ▷ Return an approximation of posterior

---

**Incremental Learning** We conducted experiments on the LeNet5 network (MNIST dataset) and the 3Conv3FC network (CIFAR-10 dataset). We applied 4 types of previously described approximations: Fine-Tuning (FT), Fully-Factorized Gaussian (FFG), Channel-Factorized Gaussian (CFG), Multiplicative Normalizing Flows (MNF). Dataset is devided into $T$ slices and perform incremenental training procedure on each of the slices consequently. We use Adam optimizer with default parameters. Optimizer state (e.g. moving moments) is reset before each incremental stage. The predictions of Bayesian models were averaged over 100 samples from the approximate posterior distribution.

**Incremental Learning with Pretraining** We use the following experiment design on the CIFAR-10 dataset. We split the dataset into two parts. First, we split the CIFAR-10 dataset into two "CIFAR-5" datasets, with 5 classes in each part (selected at random). We use the first five classes for pretraining, and then apply the incremental learning framework on the second part of the dataset. We call this task CIFAR(5+5). At the initial stage of the experiment we train a neural network on the first dataset in the conventional non-incremental setting. Then we divide the second dataset into $T$ parts and train an incremental model on each part consequently, as described in the previous sections. Such experiment design allows us to model pretraining on an unrelated task from a similar domain. We use the parameters of the network trained on one dataset to initialize the incremental training procedure on the second dataset. We use networks of the same architecture in both stages of the experiment. We use parameters of convolutional layers learned during the first stage. However, we don't use the pretrained parameters of the fully-connected layers which produce the network's predictions since we classify objects into different set of classes on different stages. It is a typical technique, as the convolutional layers tend to extract task-independent features, whereas the fully-connected layers use these features to obtain the prediction.

B    MODELS

Table 1: Full description of the LeNet-5-Caffe LeCun et al. (1998) architecture for the MNIST dataset.

| Block type | Width | Stride | Padding | Input shape | Nonlinearity |
|---|---|---|---|---|---|
| Convolution ($5 \times 5$) | 20 | 1 | 0 | $M \times 1 \times 28 \times 28$ | ReLU |
| Pooling ($2 \times 2$) | | 1 | 0 | $M \times 20 \times 24 \times 24$ | None |
| Convolution ($5 \times 5$) | 50 | 1 | 0 | $M \times 20 \times 12 \times 12$ | ReLU |
| Pooling ($2 \times 2$) | | 1 | 0 | $M \times 50 \times 8 \times 8$ | None |
| Fully-connected | 500 | | | $M \times 800$ | ReLU |
| Fully-connected | 10 | | | $M \times 500$ | Softmax |

Table 2: Full description of the small 3Conv3FC Hinton et al. (2012) architecture for the CIFAR-10 dataset.

| Block type | Width | Stride | Padding | Input shape | Nonlinearity |
|---|---|---|---|---|---|
| Convolution $(5 \times 5)$ | 32 | 1 | 2 | $M \times 3 \times 32 \times 32$ | ReLU |
| Pooling $(3 \times 3)$ | | 2 | 0 | $M \times 32 \times 32 \times 32$ | None |
| Convolution $(5 \times 5)$ | 64 | 1 | 2 | $M \times 32 \times 15 \times 15$ | ReLU |
| Pooling $(3 \times 3)$ | | 2 | 0 | $M \times 64 \times 15 \times 15$ | None |
| Convolution $(5 \times 5)$ | 128 | 1 | 1 | $M \times 64 \times 7 \times 7$ | ReLU |
| Pooling $(3 \times 3)$ | | 2 | 0 | $M \times 128 \times 7 \times 7$ | None |
| Fully-connected | 1000 | | | $M \times 1152$ | ReLU |
| Fully-connected | 1000 | | | $M \times 1000$ | ReLU |
| Fully-connected | 10 | | | $M \times 1000$ | Softmax |

## C  MNFs FOR THE INCREMENTAL LEARNING TASK

This section describes the proposed MNF-based approximation (Louizos & Welling (2017)) of the posterior distribution that is suitable for the incremental learning task, and describes the training procedure of this model. Unfortunately, we can't apply MNFs inference technique for the incremental learning setting. For the variational in incremental learning task we have to estimate $D_{\mathrm{KL}}(q(w|\phi_t) \,||\, q(w\,|\,\phi_{t-1}))$, where $q(w\,|\,\phi_{t-1})$ is the posterior approximation obtained at the previous step of incremental learning. This KL-term is intractable, moreover now we can't evaluate neither the new variational approximation $q(w\,|\,\phi)$ nor the old one $q(w\,|\,\phi_{t-1})$, as the computation of these distributions requires marginalization over the whole space of latent variables $z$. Alternative idea is to include latent variables $z$ into the original probabilistic model as a new parameter:

$$p(\mathcal{D}\,|\,w)p(w\,|\,z)p(z). \tag{7}$$

To simplify notation used in derivation we next omit parameter $\phi$ and time indexing

$$q(w) = q(w\,|\,\phi_t), \quad \widetilde{q}(w) = q(w\,|\,\phi_{t-1}), \tag{8}$$

Now our goal is to obtain the joint estimation $q(w, z) \approx p(w, z\,|\,\mathcal{D})$. Consider an arbitrary step of the incremental learning. Denote variational distribution achieved at the previous step as $\widetilde{q}(w, z) = \widetilde{q}(w\,|\,z)\widetilde{q}(z)$. Now consider a probabilistic model for the current step of the incremental learning.

$$p(\mathcal{D}, w, z) = p(\mathcal{D}\,|\,w)\widetilde{q}(w, z) \tag{9}$$

We can approximate the posterior $p(w, z\,|\,\mathcal{D}) \approx q(w, z)$ via optimization of the variational lower bound:

$$\mathcal{L} = \underbrace{\mathbb{E}_{q(w,z)} \log p(\mathcal{D}\,|\,w)}_{\text{Data term}} - \underbrace{D_{\mathrm{KL}}(q(w, z)\,||\,\widetilde{q}(w, z))}_{\text{KL-divergence}} \to \max_{q(w,z)\in\mathcal{Q}} \tag{10}$$

For simplicity we use notation as in (10). We can expand the data term and rewrite in in the following form:

$$\mathbb{E}_{q(w,Z)} \log p(\mathcal{D}\,|\,w) = \mathbb{E}_{q(z)}\mathbb{E}_{q(w\,|\,z)} \log p(\mathcal{D}\,|\,w) \simeq \log p(\mathcal{D}\,|\,\hat{w}(\hat{z})), \tag{11}$$

where

$$\hat{z} \sim q(z) \\ \hat{w}(\hat{z}) \sim q(w\,|\,\hat{z}) \tag{12}$$

For a fully-connected layer it is equivalent to the following sampling scheme:

$$\hat{z} = \mathrm{NF}(z_0), \qquad z_0 \sim q_0(z_0), \\ \hat{w}_{ij}(\hat{z}_i) = \hat{z}_i\mu_{ij} + \sigma_{ij}\varepsilon_{ij}, \qquad \varepsilon_{ij} \sim \mathcal{N}(0, 1). \tag{13}$$

Here the denotation NF stands for the Normalizing Flow. Instead of sampling weights directly, we can apply the local reparameterization trick described by Louizos & Welling (2017), which concludes the computation of the data term.

Now we need to calculate the KL divergence term. We can rewrite it in the following way:

$$-D_{\mathrm{KL}}(q(w, z)\,||\,\widetilde{q}(w, z))) = \\ = -\mathbb{E}_{q(w,z)} \log q(w, z) + \mathbb{E}_{q(w,z)} \log \widetilde{q}(w, z) \tag{14}$$

$$\mathbb{E}_{q(w,z)} \log \widetilde{q}(w,z) = \mathbb{E}_{q(z)} \mathbb{E}_{q(w\,|\,z)} \log \widetilde{q}(w\,|\,z) + \mathbb{E}_{q(z)} \log \widetilde{q}(z) \tag{15}$$

The expectation $\mathbb{E}_{q(w\,|\,z)} \log \widetilde{q}(w\,|\,z)$ is essentially a cross-entropy between two normal distributions and it can be computed analytically. The expectation $\mathbb{E}_{q(z)} \log \widetilde{q}(z)$ can be efficiently sampled, as the log-density $\log \widetilde{q}(z)$ is computed analytically as the log-density of a Normalizing Flow. Now we need to compute the expectation $-\mathbb{E}_{q(w,z)} \log q(w,z)$. It can be written in the following form:

$$-\mathbb{E}_{q(w,z)} \log q(w,z) = -\mathbb{E}_{q(z)} \mathbb{E}_{q(w\,|\,z)} \log q(w\,|\,z) - \mathbb{E}_{q(z)} \log q(z) \tag{16}$$

We can remove the expectation w.r.t. the distribution $q(z)$ in the first term, as the entropy of normal distribution $q(w_{ij}\,|\,z_i) = \mathcal{N}(w_{ij}\,|\,z_i\mu_{ij}, \sigma_{ij}^2)$ depends only on its variance $\sigma_{ij}^2$ and does not depend on $z_i$. Therefore we have

$$-\mathbb{E}_{q(w,z)} \log q(w,z) = \mathcal{H}(\mathcal{N}(w_{ij}\,|\,z_i\mu_{ij}, \sigma_{ij}^2)) - \mathbb{E}_{q(z)} \log q(z) \tag{17}$$

The first term is computed analytically and the second term can be easily sampled using the log-density, computed by the Normalizing Flow. Therefore if we use the joint model $q(w,z)$ instead of the marginalized model $q(w) = \int q(w\,|\,z)q(z)dz$, we do not need to perform the nested variational inference procedure.

The only things left are to write down the cross-entropy of two normal distributions $q(w\,|\,z)$ and $\widetilde{q}(w\,|\,z)$, the entropy of $q(w\,|\,z)$ and sampling procedures for $\mathbb{E}_{q(z)} \log q(z)$ and $\mathbb{E}_{q(z)} \log \widetilde{q}(z)$.

Sampling for $q(z)$ ("$\simeq$" denotes unbiased estimation):

$$\hat{z} \sim q(z) \Leftrightarrow \hat{z} = \mathrm{NF}(z_0), \ z_0 \sim q_0(z_0) \tag{18}$$

$$\mathbb{E}_{q(z)} \log q(z) \simeq \log q(\hat{z}) \tag{19}$$

$$\mathbb{E}_{q(z)} \log \widetilde{q}(z) \simeq \log \widetilde{q}(\hat{z}) \tag{20}$$

These estimators can be differentiated w.r.t. the Normalization Flow parameters to obtain an unbiased gradient estimate.

Cross-entropy of $q(w\,|\,z)$ and $\widetilde{q}(w\,|\,z)$:

$$-\mathbb{E}_{q(w_{ij}\,|\,z_i)} \log \widetilde{q}(w_{ij}\,|\,z_i) = \frac{1}{2} \log 2\pi\widetilde{\sigma}_{ij}^2 + \frac{\sigma_{ij}^2 + z_i^2(\mu_{ij} - \widetilde{\mu}_{ij})^2}{2\widetilde{\sigma}_{ij}^2} \tag{21}$$

Entropy of $q(w\,|\,z)$:

$$\mathcal{H}(\mathcal{N}(w_{ij}\,|\,z_i\mu_{ij}, \sigma_{ij}^2)) = \frac{1}{2} \log 2\pi\sigma_{ij}^2 e \tag{22}$$

This concludes the definition of the MNF approximation for the Bayesian incremental learning procedure.

DISCUSSION OF THE MODEL

Multiplicative normalizing flows provide us a good approximation of the posterior distribution and show high predictive performance in practice. Despite it, the multiplicative normalizing flows contain a lot of parameters, which means that they are slow and can cause problems with optimization.

