# OpenReview forum: "Bayesian Incremental Learning for Deep Neural Networks"
_ICLR.cc/2018/Workshop — Accept_

### Official Review · AnonReviewer3 · 2018-02-23
**Interesting**

**Rating:** 8
**Confidence:** 5

**Review:**

The idea of applying Bayesian updates to effect online learning is not new. However, this paper provides interesting insights regarding the technical implementation details that are needed for applying this concept to deep networks.

---

### Official Review · AnonReviewer1 · 2018-03-10
**Simple but practical idea, systematically tested.**

**Rating:** 7
**Confidence:** 3

**Review:**

In this work, Bayesian online updating is used to update the weights of a DNN as more training data becomes available. To avoid the direct intractability, a variational lower bound is used to approximately compute the updated weight densities. Different approximating distributions for the posterior are considered. Pretraining is framed as using a prior in this approach.

The paper is clearly written and systematically tests several potential approximating distributions, instead of simply considering mean field. The experiments show the clear advantage of the proposed approach with respect to fine-tuning, as well as the influence in accuracy of different posterior families.

Not a very novel approach, but targeting a useful use case and well-executed evaluation.

---

### Official Review · AnonReviewer2 · 2018-03-12
**Straightforward approach but informative experimental results**

**Rating:** 6
**Confidence:** 4

**Review:**

This paper proposed incremental learning based on the Bayesian framework which inherently has the incremental update.
The approach is straightforward but the experimental results seem to be interesting and informative for ICLR workshop.
They may lead to the next important step for Bayesian deep learning.

---

### Public Comment · ~Thang_D_Bui1 · 2018-03-29
**Variational continual learning (arxiv 2017, ICLR 2018)**

We enjoyed reading your paper. However, we believe that the relationship to Variational Continual Learning [1] should be made explicit. Variational Continual Learning uses the same general setup: a combination of online variational inference and Monte Carlo variational inference employing the reparameterization trick.

[1] Cuong V. Nguyen, Yingzhen Li, Thang D. Bui, and Richard E. Turner. Variational continual learning, 2017.

Full disclosure: we are the authors of this paper.

---

### Decision · Program_Chairs · 2018-03-20
**ICLR 2018 Workshop Acceptance Decision**

**Decision:**

Accept

**Comment:**

Congratulations, your paper was accepted to the ICLR workshop.